# Study of a Waste Kaolin as Raw Material for Mullite Ceramics and Mullite Refractories by Reaction Sintering

**DOI:** 10.3390/ma15020583

**Published:** 2022-01-13

**Authors:** Pedro José Sánchez-Soto, Dolores Eliche-Quesada, Sergio Martínez-Martínez, Luis Pérez-Villarejo, Eduardo Garzón

**Affiliations:** 1Institute of Materials Science of Sevilla (ICMS), Joint Center of the Spanish National Research Council (CSIC)-University of Sevilla, 41092 Sevilla, Spain; pedroji@icmse.csic.es (P.J.S.-S.); smartine@ujaen.es (S.M.-M.); 2Department of Chemical, Environmental and Materials Engineering, University of Jaén, Campus Las Lagunillas s/n, 23071 Jaén, Spain; deliche@ujaen.es (D.E.-Q.); lperezvi@ujaen.es (L.P.-V.); 3Center for Advanced Studies in Earth Sciences, Energy and Environment (CEACTEMA), University of Jaén, Campus Las Lagunillas s/n, 23071 Jaén, Spain; 4Department of Engineering, University of Almería, La Cañada de San Urbano s/n, 04120 Almería, Spain

**Keywords:** kaolin, raw materials, kaolinite, washing, mullite, ceramics, refractories, reaction sintering, α-alumina

## Abstract

A deposit of raw kaolin, located in West Andalusia (Spain), was studied in this work using a representative sample. The methods of characterization were X-ray diffraction (XRD), X-ray fluorescence (XRF), particle size analysis by sieving and sedimentation, and thermal analysis. The ceramic properties were determined. A sample of commercial kaolin from Burela (Lugo, Spain), with applications in the ceramic industry, was used in some determinations for comparison purposes. The kaolin deposit has been produced by alteration of feldspar-rich rocks. This raw kaolin was applied as an additive in local manufactures of ceramics and refractories. However, there is not previous studies concerning its characteristics and firing properties. Thus, the meaning of this investigation was to conduct a scientific study on this subject and to evaluate the possibilities of application. The raw kaolin was washed for the beneficiation of the rock using water to increase the kaolinite content of the resultant material. The results indicated that the kaolinite content of the raw material was 20 wt % as determined by XRD, showing ~23 wt % of particles lower than 63 µm. The kaolinite content of the fraction lower than 63 µm was 50 wt %. Thus, an improvement of the kaolinite content of this raw kaolin was produced by wet separation. However, the kaolin was considered as a waste kaolin, with microcline, muscovite and quartz identified by XRD. Thermal analyses by Thermo-Dilatometry (TD), Differential Thermal Analysis (DTA) and Thermo-Gravimetry (TG) allowed observe kaolinite thermal decomposition, quartz phase transition and sintering effects. Pressed samples of this raw kaolin, the fraction lower than 63 µm obtained by water washing and the raw kaolin ground using a hammer mill were fired at several temperatures in the range 1000–1500 °C for 2 h. The ceramic properties of all these samples were determined and compared. The results showed the progressive linear firing shrinkage by sintering in these samples, with a maximum value of ~9% in the fraction lower than 63 µm. In general, water absorption capacity of the fired samples showed a decrease from ~18–20% at 1050 °C up to almost zero after firing at 1300 °C, followed by an increase of the experimental values. The open porosity was almost zero after firing at 1350 °C for 2 h and the bulk density reached a maximum value of 2.40 g/cm^3^ as observed in the ground raw kaolin sample. The XRD examination of fired samples indicated that they are composed by mullite, from kaolinite thermal decomposition, and quartz, present in the raw sample, as main crystalline phases besides a vitreous phase. Fully-densified or vitrified materials were obtained by firing at 1300–1350 °C for 2 h. In a second step of this research, it was examined the promising application of the previous study to increase the amount of mullite by incorporation of alumina (α-alumina) to this kaolin sample. Firing of mixtures, prepared using this kaolin and α-alumina under wet processing conditions, produced the increase of mullite in relative proportion by reaction sintering at temperatures higher than 1500 °C for 2 h. Consequently, a mullite refractory can be prepared using this kaolin. This processing of high-alumina refractories is favoured by a previous size separation, which increases the kaolinite content, or better a grinding treatment of the raw kaolin.

## 1. Introduction

Kaolinite is a dioctahedral phyllosilicate with a theoretical formula Al_2_O_3_·2SiO_2_·2H_2_O or Al_2_Si_2_O_5_(OH)_4_ with a 1:1 layered structure of [(Si_2_O_5_)]^2−^ and [Al_2_(OH)_4_]^2+^ molecular sheets and composition of 39.5 wt % Al_2_O_3_, 46.5 wt % SiO_2_ and 14 wt % H_2_O [1,2,3,4]. According to Murray [5], kaolinite can be formed by hydrothermal alteration or weathering reactions or by a combination of both processes. However, Galán and Ferrell [6] indicated that kaolinite can be found also by diagenetic processes. Clay minerals, as kaolinite, are formed by decomposition of feldspathic rock via geological processes [3]. On the other hand, feldspars are the most widespread mineral group in the world forming around 60% of the earth’s crust [7,8]. Murray and Keller [9] suggested a typical reaction sequence, using potassium feldspar (KAlSi_3_O_8_), as follows:2KAlSi_3_O_8_ + 3H_2_O → Al_2_Si_2_O_5_(OH)_4_ + 4SiO_2_ + 2KOH(Reaction 1)

It has been mentioned that if potassium ions are not properly removed (following the weathering process), illitic clays (2:1 layered structure silicates) are formed instead of kaolinite [3]. Kaolin ores, with a relative high proportion of kaolinite, are a very interesting raw materials, with important and wide applications due to the properties of kaolinite layer silicate: low surface area (compared to other clay minerals, such as smectites), layered appearance, white or almost white colour, chemically inert in a wide range of pH (4–9), with low heat and electricity conductivity, refractoriness (high-temperature resistance of ~1550 °C) and non-abrasive [1,3,9,10,11,12,13,14]. Consequently, powdered and processed raw kaolin materials with a relatively high-proportion of kaolinite can be technologically applied in several industries, such as porcelain manufacture, ceramics, refractories, paper, paints, composite materials, polymers, and many more [1,3,9,10,11,12,13,14].

On the other hand, thermal decomposition of kaolinite 2SiO_2_·Al_2_O_3_·H_2_O produces the aluminosilicate named “mullite” with composition 3Al_2_O_3_·2SiO_2_ in high relative proportion, with a segregation of amorphous silica which crystallizes as cristobalite by heating [1,2,3,15,16,17,18,19]. The summary reaction is as follows:3(Al_2_O_3_·2SiO_2_·2H_2_O) → 3Al_2_O_3_·2SiO_2_ + 4SiO_2_ + 6H_2_O (Reaction 2)

It is emphasized that mullite is a rare mineral in nature, found only as a very scarce crystallized mineral at Mull island (Scotland, United Kingdom) [15]. It is the origin of the mineral name of “mullite”. The nominal composition of mullite is 3Al_2_O_3_·2SiO_2_ with 72 wt % Al_2_O_3_ and 28 wt % SiO_2_ and it must be obtained by synthesis because mullite ores are not available. Mullite is one of the most common crystalline phases present in ceramics, for instance porcelains and refractories [1,2,3,10,11,12,13,14,15,16,17,18,19]. According to the Al_2_O_3_-SiO_2_ binary equilibrium phase diagram, mullite is the only stable crystalline phase at normal pressure with high melting point (1830 °C) [1,15,20,21]. Consequently, a complete chemical reaction to produce mullite (or *mullitization reaction*) from 39.8 wt % Al_2_O_3_ (kaolinite) to reach 72 wt % Al_2_O_3_ (mullite) will require an addition of alumina (or precursors of alumina). This alumina will react with segregated silica from heated kaolinite (or Al-Si precursor) yielding mullite. Thus, it can be achieved the stoichiometric 3:2 mullite molar ratio with high melting point at normal pressure (1830 °C) [1,20,21]. It has been extensively reported in the scientific literature the preparation of mullite materials by this procedure of synthesis using inorganic precursors (kaolinite combined with alumina) with enough purity at relatively low cost [1,12,15,22,23,24,25,26,27,28,29,30,31,32].

The materials containing high proportion of mullite show excellent properties, such as good chemical and thermal stability, high melting point (1830 °C), stability in oxidative atmospheres, high-temperature strength, creep resistance, high-resistance to thermal shock, good fracture strength (~200 MPa), appropriate toughness (~2.5 MPa·m^1/2^), low thermal conductivity (~6 kcal·m^−1^·h^−1^·°C^−1^ at 20 °C), low density (3.17 g/cm^3^), low expansion coefficient (~4.5 × 10^−6^ K^−1^), low dielectric constant, very high-transmittance in the mid-IR range and retention of mechanical properties to elevated temperatures [1,12,13,14,15,22,23,24,25,26,27,28,29,30,31,32]. Due to these properties, mullite is applied as structural advanced materials for high-temperature engineering applications. Traditional and advanced materials based on high-content of mullite are very important as refractory ceramics with relevant electronical, catalytical (as substrate) and optical applications [15,23,24,27,28,29,30]. For instance, high-technical mullite-based refractories are valuable to use as kiln furniture and lining of high-temperature furnaces. Several important industries use this kind of refractories, for instance iron and steel, cement, catalysts, petrochemicals, glass and many other advanced functional applications [20,21,22,23,24,25,26,27,28,29,30,31,32]. Thus, it is very important and a real problem the valorization of mining residues by the production of mullite-based ceramic materials [32,33,34,35,36,37,38,39,40,41]. This investigation is a contribution for this purpose.

On the other hand, it is important to remark that the ceramic industry has an old tradition in the Spanish region of Andalusia [42,43]. There are important deposits of common clays, but it is a problem that some kaolin deposits located in this region have not been studied systematically in relation to mineralogy, chemistry and technological properties. First attempts were performed a few years ago as a contribution to the knowledge of Spanish kaolin deposits and their possible applications in the ceramic industry [42,43,44]. A kaolin deposit located in West Andalusia is studied in the present investigation. This kaolin derived from volcanic rocks, according to earlier studies [44,45,46], within the volcano-sedimentary complex of the Iberian Pyrite Belt [45]. The first scientific studies on samples of this kaolin deposit proposed that it belongs to the alteration products of volcanic intrusive rocks (volcanic-sedimentary system) constituted by kaolinite, (meta)halloysite and alunite with illite, being in contact with carboniferous shales [44]. This kaolin has been produced by alteration of feldspar-rich rocks and originated a mixture of kaolinite and (meta)halloysite with illite, as pointed out above. Quartz and feldspars are the main mineral impurities. The proportion of (meta)halloysite and the crystallinity of kaolinite increases with depth in the deposit [44]. It is known from years that this raw kaolin was applied in local manufactures of ceramics and refractories as an additive, but without studies concerning its beneficiation, potential and properties as raw ceramic material. It is a true problem to be solved with investigations, such as the presented here.

In the present work, this kaolin ore has been considered as a matter of investigation and new results are reported on this subject. The main objective was to conduct a scientific study on the characterization of the raw kaolin and its beneficiation and the possibility of preparation of a product with mullite in larger relative proportion using the raw kaolin material.

## 2. Materials, Experimental Procedures and Techniques

### 2.1. Materials and Experimental Procedures

A selected raw sample, representative of the kaolin deposit, was considered after an in situ examination (photographs 1 and 2 included in Appendix A). The sample was air-dried for 24 h. Then, it was wet sieved in a single batch of 5 kg to pass several stainless steel sieves, starting from 10,000 µm and being the last one 63 µm. A polyethylene tank of 60 L was used. First of all, the sample was treated using 10 L of deionized water and the particles disaggregated using intensive stirring (Selecta equipment). Next, more water was slowly added up to 50 L and a new operation of stirring. After 3 h of intensive stirring, the slurry was pumped towards a tower arranged with several stainless-steel sieves starting from 10,000 µm (Figure 1). The resultant wet powders retained in each sieve were washed using deionized water. The washed kaolin, considered the fraction under 63 µm, was decanted in a second polyethylene tank of 60 L and the wet powder (fine fraction or fraction under 63 µm) was collected. All the fractioned samples of the raw kaolin were air-dried overnight and dried at 110 °C/24 h. After that, they were weighed to record the sized results (Figure 1) and stored in plastic bags.

An amount of the raw kaolin sample investigated in this work was dried during 24 h and dry milled using a hammer mill (Verdés, Barcelona, Spain) with a sieve of 1000 µm outside the mill.

A sample of commercial kaolin from Burela (Lugo, Spain) has been used in some determinations for comparison purposes. The sample is a washed kaolin, with 85 wt % kaolinite, purchased to ECERSA and applied in the ceramic industry.

It was used as raw material for the present study a powdered commercial corundum (α-Al_2_O_3_) sample, with purity 98.9 wt % of Al_2_O_3_, being a raw material applied in the fabrication of high-alumina refractories. It was supplied by a Company (Alfran Refractories, Alcalá de Guadaira, Sevilla, Spain). This sample contains 0.55 wt % Na_2_O, 0.08 wt % Fe_2_O_3_, 0.25 wt % TiO_2_ and shows 0.16 wt % of LOI (“loss on ignition”). After ball-milling this powder, the resultant material shows 90 wt % of particle size under 63 µm. This sample was used to the preparation of high-mullite ceramic bodies using the kaolin sample studied in this investigation as raw material.

The preparation of high-mullite materials involved the powdered α-Al_2_O_3_ prepared as described above and the milled kaolin (using the hammer mill). Both raw materials were mixed in order to achieve the required mixture to yield the stoichiometry composition of mullite (3Al_2_O_3_·2SiO_2_) or Al_2_O_3_/SiO_2_ = 1.5 molar ratio. For this purpose, separate aqueous suspensions, using deionized water, of α-Al_2_O_3_ and the milled kaolin were obtained. The pH was adjusted by addition of HCl (35 vol. %). Both suspensions were mixed and after vigorous stirring during 3 h, the wet solid product was dried at 110 °C. Finally, the powder was subjected to an additional ball milling treatment for 1 h to destroy agglomerates and finally sieved under 63 µm. The resultant product showed a residue lower than 1 wt % using this sieve.

### 2.2. Techniques

Chemical analysis of selected samples was performed by X-ray Fluorescence Analysis (XRF). Ground powders were prepared and after that, pressed pellets were obtained. Boric acid was added to prepare these pressed pellets. A Siemens SRS-3000 sequential XRF spectrometer, Siemens, Aktiengesellschaft, Karlsruhe, Germany, with standard Rh X-ray tube, was the equipment used in the measurements. Quality control of these XRF analysis was carried out by calibration curves with standard certified material samples (International Association of Geologists). The errors for the major elements were found below 10%.

Particle size analysis was performed by sedimentation, based on the Stokes’ law following the method and set up proposed by the Spanish Group of Kaolins [45]. As a chemical dispersant, it was used an aqueous solution of sodium hexametaphosphate.

Mineralogical phase analysis of samples was carried out using X-ray powder diffraction (XRD). Ground samples were prepared in an agate mortar and pestle. Random preparations were scanned using an X-ray diffractometer Siemens, D-501 model, Siemens, Aktiengesellschaft, Karlsruhe, Germany, at 40 kV and 20 mA, graphite monochromator and Ni-filtered CuKα radiation. The semiquantitative phase analysis of the samples based on the XRD results was performed by the methods previously described in the literature and applied in clay mineralogy studies [45,47,48,49,50]. Selected X-ray diagnostic peaks, in general of maximum intensity, of previous identified crystalline phases were used in these determinations. The peak areas were measured taking into account several sources of error which influence the shape of XRD peaks, in particular X-ray background, possible grinding effects in the mineral phases and orientation of layer silicates [45,49]. These peak areas are divided by particular and corresponding coefficients previously determined or using data for silicates and clay minerals published by several authors [45,47,48,51]. Then, the relative percentages (in wt %) between the minerals can be obtained and compared.

Thermal behaviour and characterization of selected samples was investigated using Differential Thermal Analysis (DTA), Termogravimetric (TG) analysis and Dilatometric analysis or Thermo-dilatometry (TD). DTA-TG diagrams were obtained simultaneously using a Rigaku Thermal Analyzer, Rigaku Co., Tokyo, Japan, PTC-10A model, with Pt/Pt-Rh 13% thermocouples, and using a data processor DPS-1. For these runs, samples of 40 mg were used. They were packed in a platinum crucible and α-Al_2_O_3_ calcined at 1200 °C as a reference material for DTA in other platinum crucible. The experimental conditions were: heating rate 12 °C/min, flowing air and heating from 20 to 1200 °C as maximum.

Dilatometric analysis was carried out in air using a Malkin BCRA dilatometer, British Ceramic Research Association, Stoke-on-Trent, United Kingdom, which was adapted with an automatic programmer and recording. To perform this analysis, as a first step, it was necessary to prepare test bars using plaster of Paris moulds and wet samples with enough plasticity. The test bars were dried at 110 °C in an oven and controlled the size by a calliper. The first step was a dilatometric run performed using green samples, heated at 6 °C/min and fired up to 1000 °C for 1 min inside the furnace of the dilatometric equipment. The fired bars were cooled inside the furnace up to room temperature. They were studied by a second dilatometric analysis under the same experimental conditions. Both curves (green and fired) were obtained and compared.

The determination of ceramic properties was performed using cylindrical pellets prepared by uniaxial dry pressing of the powders at 40 MPa. An amount of ~5 wt % of deionized water was mixed to these powders before pressing. The obtained pellets were air-dried for 4 h. After that, they were treated at 60 °C for 4 h using an oven. These dried pellets were heated, in air, using a high-temperature furnace to several temperatures at a heating rate of 6 °C/min. After achieving the selected firing temperatures, 2 h of soaking time was fixed for each batch of samples. Then, the furnace was cut off and the fired pellets were cooled inside the furnace up to room temperature. The weight loss of the pellets after firing (wt %) was determined as the mass loss between drying (at 110 °C) and firing temperatures. Linear firing shrinkage (in percentage) was determined by measuring the length of the samples before and after the firing treatments. A calliper was used with a precision of ±0.01 mm. Water absorption capacity (in wt %), bulk density (g/cm^3^) and apparent or open porosity (in wt %) of the fired pellets were measured by the immersion technique, using distilled water, saturation for 24 h and boiling in distilled water for 2 h. All the measurements were performed in triplicate following the Spanish standard procedures [52]. The fired samples were studied by XRD using a portion of the pellets after grinding.

Microstructural features of selected fired samples were studied by scanning electron microscopy (SEM). The samples were coated with a thin layer of gold by sputtering using a Balzer sputtering device. The equipment used for SEM examination was a JEOL microscope, model JSM-5400, JEOL Ltd., Tokyo, Japan. To reveal the microstructure of the fired samples eliminating the glassy phase, they were examined in fresh fractures after a previous etching using a 20% HF aqueous solution for 10 min.

## 3. Results and Discussion

### 3.1. Particle Size Separation and Analysis

Figure 1 shows the variation of particle sizes of the raw sample (using 5 kg) after washing. These results, presented as histograms, indicated variable percentages of fractions lower than 10,000 μm (Figure 1A), with variations of 32 wt % in the fraction between 10,000–2000 µm and ~9–10 wt % in the fractions between 2000–1000 µm and 1000–200 µm, with lower percentages (~5.5 wt %) in the fractions between 200–63 µm. The percentage of particles lower than 63 μm was 23.06 wt %. Kaolinite, as a clay mineral, is concentrated in the finer size fractions taking into account its physical characteristics [3,5,6,7,8,9]. According to this result, the amount of washed sample is low, being associated to the presence of larger amounts of unaltered mother rock in the kaolin deposit, which contains the kaolinite clay of interest to be beneficiated. Then, the kaolin ore must be considered as a waste kaolin. The accumulative curve (Figure 1B) summarizes the variation of these percentages, indicating that the raw sample presents progressive variations of particle sizes as determined by wet sieving. In fact, there is a linear trend in the curve from 1000 to 63 µm (Figure 1B).

An analysis by sedimentation of the fraction lower than 63 µm (Figure 2) indicated percentages of ~16 wt % of fraction lower than 2 μm (or “clay fraction” [5,9]) and ~41 wt % of fraction between 6.3–2 µm (Figure 2A). These fine fractions can be considered because kaolinite would be present in relative high proportion. The accumulative curve (Figure 2B) has been compared with a washed kaolin sample from Burela (Spain) included in this research and examined under the same conditions. Precedent studies have been performed on this deposit of kaolin (the Burela kaolin deposit) [52]. It can be noted in these plots the differences in particle sizes between both samples. As expected, the percentage of particles lower than 2 µm is higher in the sample considered as a commercial washed kaolin applied in ceramics.

The particle size analysis by wet sieving was also performed using the sample obtained by milling using the hammer mill. The results are presented in Figure 3. These results, as histograms (Figure 3A), indicated that ~60 wt % of sample is lower than 63 µm, with ~34 wt % between 200–63 µm and minor percentages of higher fractions. Practically, all the raw kaolin sample as a resultant powder after milling was under 1000 µm. Thus, with the milling treatment the sample increased the amount of particles lower than 63 µm. A significative variation to decrease the higher sizes was produced. The plot of Figure 3B shows this variation as compared to the original sample (Figure 1B).

### 3.2. Mineralogical Analysis by XRD of the Kaolin Samples

Figure 4 shows the XRD mineralogical phase analysis of the raw kaolin sample and its fraction under 63 µm. Kaolinite and quartz are the main crystalline phases present in both samples, with potassium feldspar (microcline) and muscovite mica as secondary minerals. It can be seen that kaolinite X-ray patterns increases in the fraction under 63 µm. In contrast, quartz and microcline X-ray patterns decrease in that fraction and muscovite mica does not present any variation.

Table 1 includes the semiquantitative estimation of the mineral content from the XRD diagrams. It can be confirmed by these results that the washing using water of the raw sample produced a relative increase of kaolinite content from 20 to 50 wt %. However, the washed sample with this percentage of kaolinite is accompanied by quartz (20 wt %) and microcline (10 wt %) in medium proportions besides muscovite mica (20 wt %). Furthermore, muscovite mica increases slightly in the fraction under 63 µm, although this increase is between the experimental errors in the mineralogy semiquantitative estimation. Anyway, it has been verified by XRD (Figure 4) that if the kaolinite content increases by the washing treatment, as a result of the beneficiation process, the content of quartz and microcline decreases. However, the yield of finer fractions is not an income subject. In this sense, in contrast, the raw materials of the same kaolin deposit with lower kaolinite content could be considered as a waste constituted by feldspar sands because quartz and microcline are predominant mineral phases [8].

### 3.3. Chemical Analysis by XRF

Table 2 shows the chemical analysis of the raw and wet-sieved kaolin sample under 63 μm. The chemical analysis of the kaolin sample of Burela is included for comparison. All these samples present a high content of SiO_2_ (>50 wt %), associated to the kaolinite present, quartz, microcline and muscovite mica, in the range of 51.15–73.61 wt %. The percentage of SiO_2_ is higher in the raw sample, according to the presence of quartz identified by XRD (Figure 4). The Al_2_O_3_ content is medium-low, being in the range 13.40–33.51 wt % and associated to combined silicates. The content of K_2_O must correspond to microcline and muscovite present in the samples, with variable content in the range 1.78–8.55 wt %. Iron content as Fe_2_O_3_ in all the samples is lower than 2 wt %, with a minimum value of 0.63 wt % in the kaolin of Burela and maximum (1.96 wt %) in the wet sieved kaolin sample under 63 µm. Possibly, the washing of the raw kaolin concentrated the iron (as iron gel oxides) in the finer sample. The contents of the rest of elements (CaO, MgO, Na_2_O) are not relevant.

On the other hand, the values of weight loss after heating at 1000 °C (“loss on ignition” = LOI) are in the range 2.69–11.90 wt % (Table 2). It is an indication of mineralogical components present in the samples which can be dehydroxylated by thermal treatment: kaolinite and muscovite. However, quartz and microcline do not present any weight loss after thermal treatment [53]. For pure kaolinite, 13.9 wt % of structural water is lost after thermal treatment at 1000 °C [1,2,3,4]. From the results of Table 2, it can be observed that the sample with higher kaolinite content (Burela with 85 wt %), the LOI is 11.90 wt %. The sample with lower kaolinite content (20 wt %, see Table 1) is the raw kaolin, as expected, and it shows the lower LOI value (2.69 wt %, see Table 2). In fact, as observed in Table 2, the washed kaolin sample under 63 µm studied in this work shows higher percentage of LOI, being this result associated to the increase of the amount of kaolinite (Table 1).

Assuming that all the LOI is a consequence of the kaolinite content as single clay mineral, being dehydroxylated by thermal treatment, it is possible to estimate the percentage of kaolinite in the samples. Taking into account that pure kaolinite lost 13.9 wt % (i.e., the LOI of a pure kaolinite is this value) as structural water [1,2,3,4], the contents of kaolinite for the raw kaolin sample and its fraction under 63 µm are calculated as ~19 wt % and ~47 wt %, respectively (Table 2). Thus, the washing process of the raw kaolin produced an increase of the kaolinite percentage in the sample, although it was assumed in this estimation that kaolinite is the only clay mineral present. Furthermore, the same estimation applied to the washed kaolin of Burela indicates a kaolinite content of ~86 wt %. These all estimations of kaolinite content in the studied samples are in well agreement, taking into account the errors in the XRD semiquantitative estimation, with the mineralogical study performed by XRD (Table 1). In particular, the sample of kaolin under 63 µm presents a kaolinite content by XRD of 50 wt % (Table 1), being obtained ~47 wt % using the above LOI approximation (Table 2).

Furthermore, the results of Table 2 show that the molar ratio SiO_2_/Al_2_O_3_ in the raw kaolin is relatively high (9.33) but diminishes in the fraction under 63 µm (4.52), being in agreement with the kaolinite content estimated by XRD (Table 1). However, this ratio is lower (2.59) in the kaolin sample of Burela which contains ~85 wt % of kaolinite after industrial washing (Table 1).

Two parameters are of interest for an in-depth study of the kaolin samples taking into account both mineralogical and chemical characteristics [52,54,55]. It can be described as follows. For this study, it can be considered a mineralogical index alteration (MIA) and a chemical index of alteration (CIA). The first one was proposed by Nesbitt [54], being a dimensionless number between 0 and 100 calculated according to Equation (1):MIA = [Qz/(Qz + KF + Pl)] × 100 (1)
being Qz = % Quartz in the sample, KF = % potassium feldspar and Pl = % Plagioclase in the kaolin samples determined by mineralogical analysis (in wt %) by XRD.

The CIA index [52,55] is calculated using the molecular proportions as follows:CIA = [Al_2_O_3_/(Al_2_O_3_ + CaO* + Na_2_O + K_2_O)]/100(2)
being CaO* the amount of CaO combined in the silicate fraction of rock. This parameter is useful to evaluate the degree of chemical weathering focusing on feldspars. It is also a dimensionless number between 0 and 100.

Table 3 presents the results of mineralogical and chemical characteristics (MIA and CIA parameters as described above), being calculated from mineralogical (XRD) and chemical (XRF) analyses of the raw kaolin and its fraction under 63 µm (Table 1 and Table 2) and using Equations (1) and (2). The same parameters for the kaolin of Burela have been calculated and included for the sake of comparison purposes. The kaolinite content, determined by XRD (Table 1), is included in the same Table 3. The fraction under 63 µm of the raw kaolin (with 50 wt % of kaolinite) shows a MIA value of 66.66, being higher that the value of this parameter for the raw kaolin sample, with lower content of kaolinite (20 wt. %). Then, it can be deduced that this sample is a partially altered sample although it is not as the sample of kaolin of Burela, which shows a MIA value of 100.

On the other hand, the CIA values are ~60 and ~84 for the raw kaolin and its fraction under 63 µm, being higher for the sample of kaolin of Burela (~93). It is an indicator that the alteration or kaolinization is higher in this sequence, suggesting a significant removal of labile basic cations (for instance Mg, Na and K) relative to stable residual elements (Si, Al, Fe), as pointed out by Galán et al. [52]. Therefore, the loss of alkaline (and alkaline-earth) elements can be attributed to the kaolinitization of alkali-feldspars and mica, considered the minerals most prone to weathering in the kaolin deposit [5,6,7,8,9]. For a deep study, however, it would be necessary an examination of representative sampling of occurrences and associate rocks in the kaolin deposit. For instance, CIA values obtained for bulk kaolin samples of Burela range from 77.48 to 96.7 and MIA values for representative sample rocks of the Burela deposit range from 29 to 100 [52].

### 3.4. Thermal Analysis by DTA-TG and Thermal-Dilatometry

The thermal behaviour (DTA-TG) of the kaolin sample after wet washing and sieving under 63 µm is included in Figure 5. This sample contains the higher percentage of kaolinite as compared with the raw kaolin sample (Figure 4 and Table 1 and Table 2), with an estimation of 47 wt %. It contains kaolinite, muscovite mica, microcline and quartz. It should be noted that pure microcline and quartz do not show any weight loss by TG [53]. Following the progressive heating of this sample, first of all, it is observed a small endothermic DTA effect before 200 °C, with maximum peak at 85 °C. This DTA effect is associated with a low weight loss by TG. Both thermal events can be attributed to the loss by heating of mechanically-retained water molecules. Further heating produces a very small DTA effect with maximum peak at ~320 °C, being accompanied by a slight weight loss by TG in the range 250–380 °C. This effect can be explained due to the presence of iron oxide gels, which lost water by thermal treatment [56,57]. The content of Fe_2_O_3_ in this sample is 1.96 wt % (Table 2) and may be associated with iron oxide gels. The small thermal effects detected by thermal analysis have been showed and discussed above. Next, the main thermal effects will be shown and discussed in this section.

In general, the DTA-TG profiles of this sample are characteristic of pure kaolinite, with a broad endothermic DTA effect with maximum peak at 540 °C, and the main weight loss observed by TG in the temperature range 300–900 °C. This effect is associated to kaolinite dehydroxylation, with loss of structural OH groups and formation of an amorphous phase (to X-rays) named as “metakaolinite” [1,2,3,11,15,16,17,18,19]. A characteristic exothermic DTA effect is observed at ~995 °C (Figure 5) without weight loss by TG, being associated to the decomposition of metakaolinite previously formed by heating [1,2,3,11,15,16,17,18,19]. This exothermal DTA effect in pure kaolinite samples has been studied along years by many investigators with controversies [1,2]. It has been attributed to the formation of either a transient alumina-type spinel (γ-Al_2_O_3_ solid solution) or mullite nuclei or both, because at DTA heating rates (in the present case 12 °C/min) both spinel and mullite have the possibility of forming concurrently [1].

The results obtained by DTA-TG are in relation to the observed thermo-dilatometric behaviour (Figure 6). The original raw kaolin and the wet sieved fraction under 63 µm have been studied by thermo-dilatometry (TD), both in green and fired (1000 °C) states. It can be observed that the thermal profiles of all these samples are similar with some light variations. There is a continuous and rapid expansion in the TD curve associated to clay dehydroxylation in green state up to ~600 °C, when the α → β phase transformation of quartz takes place [53]. This effect was not observed by DTA (Figure 5). The expansion is slower up to a maximum of 900 °C and after this temperature, the TD curve starts to decrease by the initial process of sintering which produces shrinkage. This behaviour is accelerated in the green curve of the fraction under 63 µm, with a maximum expansion at ~850 °C. Afterwards, the sintering process is more evident because a rapid shrinkage is observed. It is clear that the mineral phases present in these samples, mainly clay minerals (muscovite and kaolinite), have been dehydroxylated by thermal heating. This process produced expansion in the TD curve up to a limit (complete dehydroxylation of silicates), when sintering effect starts.

The TD curves for these samples fired at 1000 °C shows the quartz phase transition as the more relevant feature, with a remarkable change in the slope of the straight lines of expansion in the temperature range ~550–650 °C. This effect is more intense in the original kaolin sample, being of lower intensity in the fraction under 63 µm because the quartz content is lower (Table 1).

### 3.5. Ceramic Properties

The ceramic properties have been investigated for these kaolin samples with a comparative study of the kaolin of Burela under the same experimental conditions. The evolution of linear firing shrinkage (LFS, in %) and water absorption capacity (WAC, %) as a function of firing temperature is included in Figure 7. The results obtained for the raw kaolin sample and the wet-sieved sample under 63 µm, thereafter fine fraction, are presented. It can be seen the progressive shrinkage of the raw sample and the fine fraction up to maximum values of 7.5% and 8.8%, respectively, after firing at 1300 °C. Above this temperature, LFS values decrease although it is very fast for the raw sample. WAC values follows a similar trend, with a decrease of values from ~18–20% at 1050 °C up to almost zero after firing at 1300 °C, followed by an increase of WAC values. However, the fine fraction held WAC values almost zero in a firing temperature range (estimated in 100 °C), being broader than the raw sample. This evolution is similar when the milled sample, using the hammer mill, is studied. Figure 8 shows this evolution, with maximum LFS of ~8% at 1350 °C and minimum values of CAA (~3.5–4%) in the firing range 1300–1400 °C. However, LFS results indicate that deformation in the test samples started from 1350 °C.

The ceramic properties bulk density (BD, in g/cm^3^) and apparent porosity (AP, %) have been also determined for these kaolin samples. Figure 9 shows a plot of the variation of BD and AP for the raw kaolin sample and the fine fraction as a function of firing temperature, with a soaking time of 2 h. The profiles of the curves obtained under the same experimental conditions are similar to that presented in Figure 8, with maxima values of BD and AP for these samples at different firing temperatures.

As an interesting approach to evaluate the ceramic properties, it is possible to determine the vitrification temperature (T_v_), or temperature at which the AP becomes almost zero, and the temperature of the maximum of BD (T_d_). These parameters are of interest concerning densification and sintering of ceramics taking into account the vitrification or range curves, as proposed by Norris et al. [58]. It can be estimated that T_v_ is 1300 °C for both samples, with variation in T_d_, being T_d_ = 1300 °C for the raw kaolin (original) sample and T_d_ = 1350 °C for the fine fraction.

Table 4 summarizes the experimental results of estimated T_v_ and T_d_ and the values of maximum BD for the raw kaolin sample, the fine fraction, the milled kaolin sample (using a hammer mill) and the kaolin sample of Burela. It can be observed that the higher values of T_v_ and T_d_ and BD are found in the kaolin sample of Burela (T_v_ = 1500 °C, T_d_ = 1450 °C and BD = 2.49 g/cm^3^), as expected, followed by the milled kaolin sample (T_v_ = T_d_ = 1350 °C and BD = 2.40 g/cm^3^). The fine fraction is the next sample in this sequence of decreasing values (with T_v_ = 1300 °C, T_d_ = 1350 °C and BD = 2.35 g/cm^3^) and, finally, the raw kaolin sample (T_v_ = T_d_ = 1300 °C, with BD = 2.21 g/cm^3^). According to this evolution, the milled kaolin sample reached the higher values of T_v_ and T_d_ and BD of all the samples studied in this work in comparison to the kaolin of Burela.

It should be noted that there are some variations in these parameters considering raw clays. For instance, in chlorite-muscovite (illite) clays containing quartz, it has been reported T_v_ = 1200 °C and T_d_ = 1150 °C with BD = 2.30 g/cm^3^ because these clays contain a large amount of fluxes as compared to kaolinitic clays [59]. In raw sericite clays (sericite-kaolinite and sericite-pyrophyllite-kaolinite clays), as precursors or mullite materials by heating, values of T_v_ ~ 1250 °C and BD maximum values in the range 2.41–2.52 g/cm^3^ have been reported [60]. In other raw clays (quartzitic clays), the values of T_v_ and T_d_ are in the range 1200–1250 °C with BD in the range 2.25–2.43 g/cm^3^ [61]. In general, these features are in accordance with the thermal behaviour of kaolinitic and muscovite (illite/sericite)-kaolinitic clays containing quartz [10,14].

### 3.6. Development and Evolution of Crystalline Phases by Thermal Treatment

Figure 10 shows the XRD diagrams of the fraction under 63 µm of the kaolin (or washed kaolin sample, see Section 3.1) after firing in the temperature range 1000–1300 °C for 2 h. It can be detected at 1000 °C and, at 1100 °C, quartz, dehydroxylated mica and feldspar (microcline), as crystalline phases, and an amorphous phase according to the background of the XRD diagram. Amorphous silica was produced during metakaolinite decomposition (Figure 5 DTA-TG), being larger in this sample (kaolinite content ~47 wt %, see Table 2). It can give rise to a vitreous phase formation. Important changes in the evolution of the crystalline phases were observed: (1) from 1000 °C to 1100 °C, the dehydroxylated muscovite mica disappeared by thermal decomposition, and (2) from 1200 °C to 1300 °C took place the disappearance of all the feldspar (microcline) X-ray peaks. It was associated to the melting of this phase. At the same time, an increase of the X-ray background (increased hump of the X-ray diagram), associated to an amorphous or vitreous phase, was detected. 

The observation of mullite (3Al_2_O_3_·2SiO_2_) X-ray patterns was detected at 1150 °C, being formed above 1100 °C when dehydroxylated mica and metakaolinite are decomposed. There is an increase in intensity of these mullite X-ray patterns as increasing firing temperature (1200 and 1300 °C). At 1300 °C, quartz and mullite are the only crystalline phases present in the fired sample with detection of a vitreous or glassy phase. This result is in connection with the ceramic properties of the sample presented in Figure 7, Figure 8 and Figure 9 concerning the firing behaviour with the vitrification features (see Section 3.5). Then, ceramic bodies with high relative proportion of crystalline mullite and minor of quartz, besides vitreous phase, can be obtained by firing at 1300–1350 °C. In general, these thermal features are in agreement with previous results reported in studies on kaolinitic clays [19,34,39,56,60].

It should be noted that the equilibrium phase diagram K_2_O-SiO_2_-Al_2_O_3_ [62] indicates that the formation of liquid starts at 980 °C and a peritectic liquid at temperatures above 1140 °C. The relative proportion of liquid, forming the vitreous phase after cooling the samples, increases with increasing temperature. The melting of potassium feldspar was reported in the temperature range 1118–1160 °C [52], being in accordance with the present results (Figure 10). Martin-Marquez et al. [63] reported a complete melting of feldspars (microcline and albite) used in the preparation of a porcelain stoneware at 1200 °C. It is important to remark that Dondi [8] suggested that the primary function of feldspathic fluxes is to melt during firing. Therefore, a liquid phase from feldspar and muscovite was provided in the present study that is responsible for viscous flow sintering and partial vitrification. The presence of a potassium-rich phase (Table 2) in the fired kaolin sample accelerates the vitrification process due to this fluxing effect.

### 3.7. Incorporation of α-Alumina to the Kaolin Sample to Yield a Mullite Refractory Material

According to the above results, it was considered that this kaolin sample could be useful to obtain high-mullite refractory materials by addition of alumina to react with excess of silica, segregated by kaolinite thermal decomposition forming metakaolinite, according to the reaction (Reaction 3):3(Al_2_O_3_·2SiO_2_·2H_2_O) → 3Al_2_O_3_·2SiO_2_ + 4SiO_2_ (amorphous) + 6H_2_O(Reaction 3)

The formation of primary mullite from kaolinite thermal decomposition can be detected by XRD from 1150 °C (Figure 10). Further heating can produce additional mullite, considered secondary mullite, by reaction between segregated silica from kaolinite and alumina previously mixed with kaolinite:3Al_2_O_3_ + 2SiO_2_ → 3Al_2_O_3_·2SiO_2_
(Reaction 4)

For this purpose, powdered α-Al_2_O_3_ was added to the milled kaolin (using the hammer mill) in order to achieve the required mixture to yield the stoichiometry composition of mullite (3Al_2_O_3_·2SiO_2_) or Al_2_O_3_/SiO_2_ = 1.5 molar ratio, following the procedure described in experimental. The application of the milled kaolin is based in the utilization of the whole sample without size separation by washing because the yield of fraction lower than 63 µm (which contains 50 wt % of kaolinite) is relatively low (~23 wt %).

Figure 11 shows the XRD diagrams of these mixtures after firing in the temperature range 1200–1600 °C for 2 h. The XRD study showed the disappearance of feldspar (microcline) above 1200 °C and quartz above 1400 °C. Primary mullite, from metakaolinite thermal decomposition, is detected by XRD at 1200 °C although the previous result depicted in Figure 10 demonstrated that it was at 1150 °C. It can be observed in the XRD diagrams of Figure 11 the disappearance of α-Al_2_O_3_ by Reaction 4 above 1500 °C, with some relicts of quartz as residual mineral phase, and the development of mullite phase. The samples sintered at 1550–1600 °C consisted of mullite, as the main crystalline phase, and the presence of unreacted α-Al_2_O_3_. However, complete mullitization (i.e., mullite as a single phase) was not achieved under the present experimental conditions. It must be examined, for a complete study, the degree of mixing, differences in particle size and firing conditions at 1600 °C for 2 h. Thus, under the present experimental conditions, a high-mullite-(α)-alumina refractory can be obtained.

The particles of α-Al_2_O_3_ are relatively larger than kaolinite in the prepared mixture. They can be considered as an inert phase in the ground mixture up to 1350–1400 °C, when the Reaction 4 starts. It should be noted that in a previous work, Liu et al. [29,30] reported that the reaction of α-Al_2_O_3_ with kaolinite (washed kaolin sample from Caobar, Guadalajara, Spain) is initiated at about 1250 °C, being quite extensive above 1380 °C. This reaction is extremely fast at 1600 °C and above. It is indicating a strong effect of the eutectic liquid formation at ~1587 °C which is produced in the binary system SiO_2_-Al_2_O_3_ [20] although with a lower temperature if fluxes are present. In fact, the K_2_O-Al_2_O_3_-SiO_2_ system must be considered, in which the formation of a peritectic liquid at temperatures above 1140 °C is achieved [62].

In the present case it can be observed a decrease in intensity of α-alumina peaks (Figure 11) associated to the effect and the presence of fluxes (Table 2), in particular K_2_O. According to the literature, there is evidence of liquid phase sintering in the system kaolinite-alumina taking into account the characteristics of the glassy phase and the rapid kinetics of secondary mullite formation [29,30]. These previous results suggest a mechanism of solution of α-Al_2_O_3_ and precipitation of secondary mullite via a transitory liquid phase [12,26,37]. Therefore, the solution-precipitation mechanism is through the glassy phase [29,30]. It should be emphasized that the effect of glassy or vitreous phase on mullite and mullite-based ceramic composites prepared from kaolinite, sericite clays and kaolin wastes as by-products of mining has been studied by Sánchez-Soto et al. [39]. The total content of oxides (see Table 2), distinct of Al_2_O_3_ and SiO_2_, is summed as K_2_O treating this oxide as the only impurity (oxides as “K_2_O” in wt %, obtained from the chemical analysis on a calcined basis). This estimation indicated a “K_2_O content” = 4.46 wt %. The maximum “K_2_O content” in similar materials was reported as 5.74 wt % [39]. According to Chen et al. [12], in a previous study on the preparation of mullite by reaction sintering of kaolinite and alumina, the addition of Al_2_O_3_ can reduce the amount of glassy phase and, therefore, increases the amount of mullite.

### 3.8. Microstructural Observations by SEM of Thermally Treated Kaolin Samples

The SEM observation of the sample of washed kaolin under 63 µm and fired at 1300 °C (Figure 12) shows a compacted microstructure with mullite crystals. They are originated from kaolinite decomposition by firing present in the sample (~50 wt % by XRD, see Table 1), and ~47 wt % on the basis of the LOI result (Table 2).

It can be observed in detail needle shape of small sizes of mullite crystals, besides some relicts of glassy phase and quartz crystals, as expected from previous XRD results (Figure 4), when microcline has disappeared (Figure 10). These features are similar to those reported in previous papers [3,10,15,18,19,37,60].

## 4. Conclusions

The particle size separation of a representative sample of the kaolin deposit studied in this investigation was performed by wet-sieving and after that, an analysis by sedimentation of the fraction under 63 µm. The results allow conclude: (a) the presence of variable percentages of fractions lower than 2000 µm; (b) percentages of ~23 wt % of particles under 63 µm, and (c) percentages of ~16 wt % of fraction lower than 2 µm. The kaolinite content of the raw kaolin and the fraction under 63 µm were 20 and 50 wt %, respectively, as determined by XRD. It is accompanied with quartz (20–30 wt %), potassium feldspar or microcline (10–35 wt %) and muscovite (15–20 wt %) as mineral phases identified by XRD. This result is in agreement with the relative high content of silica (62–74 wt %) and alumina (13–23 wt %) determined by XRF. The content of K_2_O was 4.15–8.55 wt % in the raw and fraction under 63 µm, respectively, being associated to the presence of microcline and muscovite. According to these results, the amount of washed kaolin sample is low because large amounts of unaltered kaolin mother rock (composed by potassium and calcium feldspars, kaolinite, mica and quartz) are present. Consequently, it is considered that the kaolin studied in this work is a waste kaolin.

The results of thermal analysis by TD of the raw kaolin and the fraction lower than 63 μm showed the thermal expansion by firing up to a maximum at ~850–900 °C and the decrease of this expansion by shrinkage produced by sintering of the samples. Additional thermal analysis by DTA-TG allowed observe the kaolinitic character of the fraction lower than 63 µm (with 50 wt % of kaolinite).

The ceramic properties of uniaxially pressed and fired samples (1000–1500 °C with 2 h of soaking) showed the progressive thermal shrinkage by sintering, with water absorption and open porosity almost zero at ~1350 °C. At 1350 °C the bulk density reached maximum values of 2.21, 2.35 and 2.40 g/cm^3^ for the raw kaolin, the fraction lower than 63 µm and the milling sample, respectively. This sequence of values reflects an effect of particle size reduction, with a maximum value of 2.49 g/cm^3^ for the kaolin of Burela used for comparison. Additional parameters of interest, according to previous research [58], have been deduced from the present results: T_v_ = 1300 °C and T_d_ = 1350 °C as maximum, with slight variations considering the raw kaolin (~20 wt % of kaolinite), the fraction lower than 63 μm (~50 wt % kaolinite) and the milled kaolin sample. The kaolin of Burela (~86 wt % of kaolinite), under the same experimental conditions, showed values of T_v_ = 1500 °C and T_d_ = 1450 °C. From these results, it is concluded that the mineralogy, chemical composition and particle size influenced this behaviour.

The main changes in phase evolution have been observed by XRD from 1200 to 1300 °C, with disappearance of microcline by melting. It was in accordance with the qualitative predictions of the equilibrium phase diagram K_2_O-Al_2_O_3_-SiO_2_. The development of mullite crystals was evidenced by XRD. At 1200 °C, quartz and mullite have been identified in the fired materials besides the presence of vitreous phase. Mullite is formed by decomposition of kaolinite. The observation of mullite forming characteristic needle shape crystals was revealed by SEM. According to the present results, the formation of fully-densified and vitrified mullite materials by firing treatments was demonstrated.

As a next step of this investigation, it was examined the incorporation of α-alumina to this kaolin sample and treatments of uniaxial pressing and firing. Firing of these samples produced the increase of mullite relative proportion by reaction sintering at temperatures higher than 1500 °C for 2 h, with disappearance of quartz as demonstrated by XRD examination. Thus, a mullite refractory material can be prepared using this kaolin sample as raw material, which is favoured by a previous size separation, or better a milling treatment of the raw kaolinite.

Further studies will be considered using this kaolin as raw material, taking into account the present results—for instance, the evaluation of mechanical strength properties.

## Figures and Tables

**Figure 1 materials-15-00583-f001:**
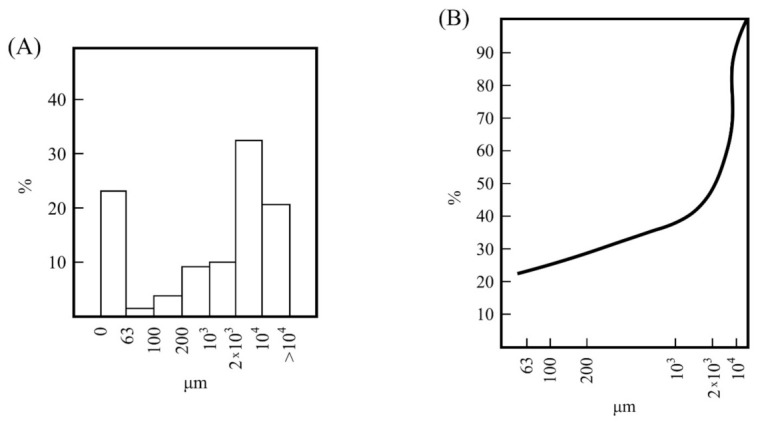
Variation of particle sizes of the raw kaolin sample: (**A**) histogram of wet-sieving separation and (**B**) curve of particle size distribution obtained from the above results.

**Figure 2 materials-15-00583-f002:**
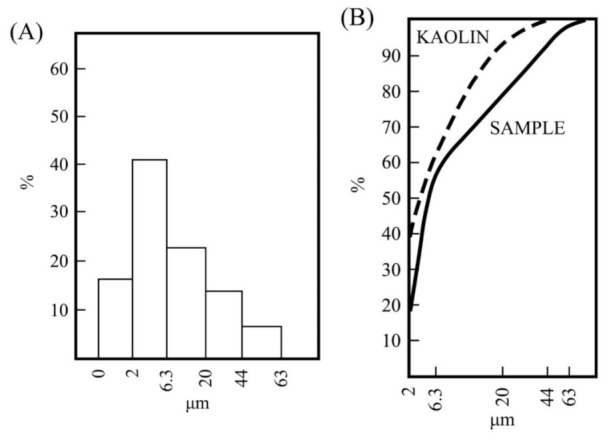
Particle size analysis of the fraction lower than 63 µm separated by wet sieving from the raw kaolin sample: (**A**) histogram of the obtained fractions and (**B**) curve of particle size distribution (labelled as SAMPLE) obtained from the above results. A washed sample of kaolin of Burela was analysed under the same experimental conditions (see the text) and the experimental curve is included (labelled as KAOLIN).

**Figure 3 materials-15-00583-f003:**
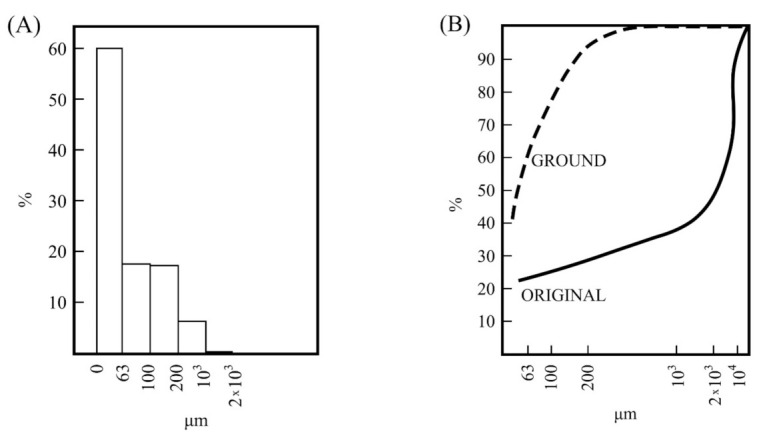
Particle size analysis of the kaolin sample studied in this work after grinding using a hammer mill: (**A**) histograms of wet-sieving analysis and (**B**) curve of particle size distribution obtained from the above results (sample labelled as GROUND). For comparison, plotted the obtained results have been plotted for the raw (unground) original kaolin sample as presented in Figure 1 (labelled as ORIGINAL).

**Figure 4 materials-15-00583-f004:**
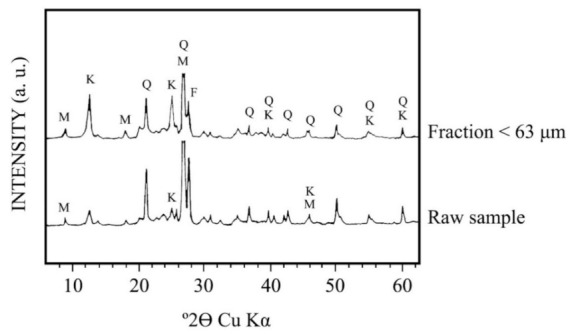
Mineralogical composition by XRD of the raw kaolin sample and fraction < 63 μm obtained by wet sieving. M = muscovite; Q = quartz; F = feldspar (microcline); K = kaolinite.

**Figure 5 materials-15-00583-f005:**
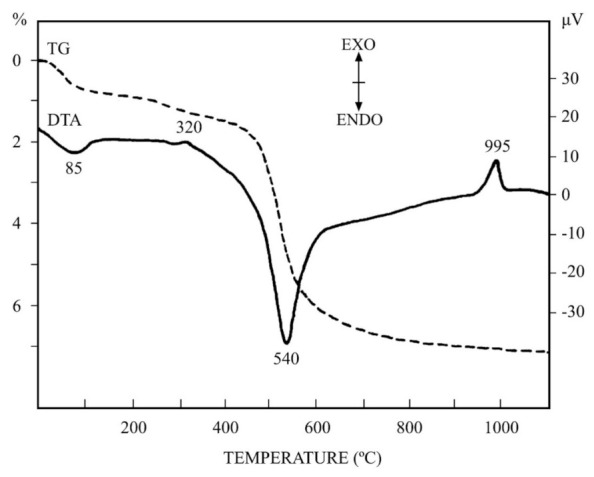
DTA-TG diagrams of the fraction under 63 μm obtained by wet sieving (heating rate 12 °C/min).

**Figure 6 materials-15-00583-f006:**
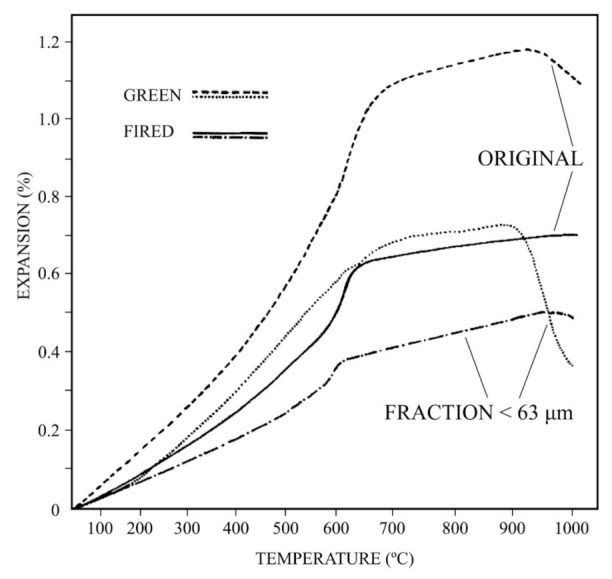
Thermal analysis results by dilatometry (thermal-dilatometry) in green and fired (1000 °C) conditions (heating rate 6 °C/min) of the raw kaolin (labelled as ORIGINAL) and fraction under 63 μm obtained by wet sieving.

**Figure 7 materials-15-00583-f007:**
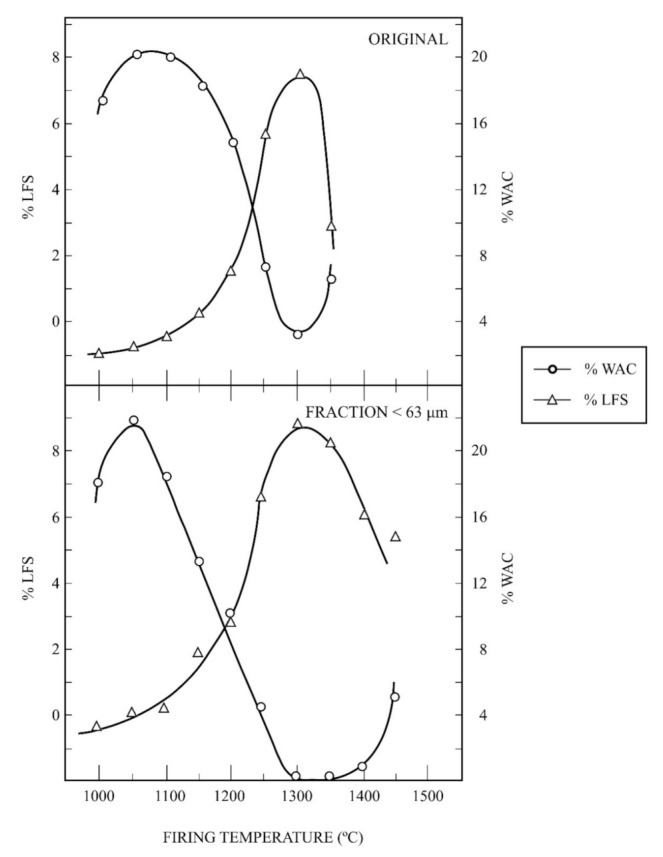
Thermal behaviour of the raw kaolin sample (labelled as ORIGINAL) and fraction under 63 μm: plot of the variation of linear firing shrinkage (LFS) and water absorption capacity (WAC) as a function of firing temperature.

**Figure 8 materials-15-00583-f008:**
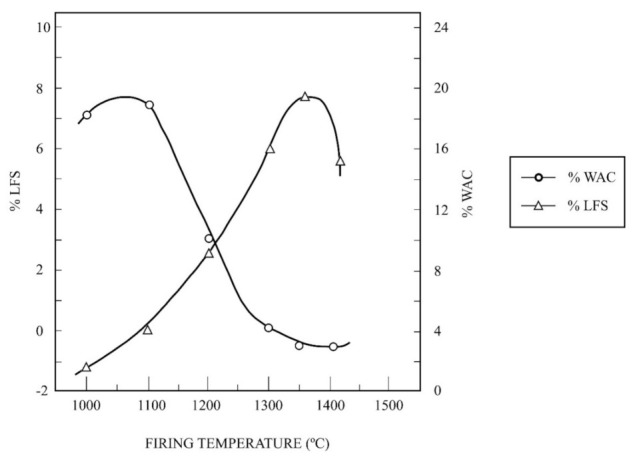
Thermal behaviour of the kaolin sample after grinding treatment using a hammer mill: plot of the variation of linear firing shrinkage (LFS) and water absorption capacity (WAC) as a function of firing temperature.

**Figure 9 materials-15-00583-f009:**
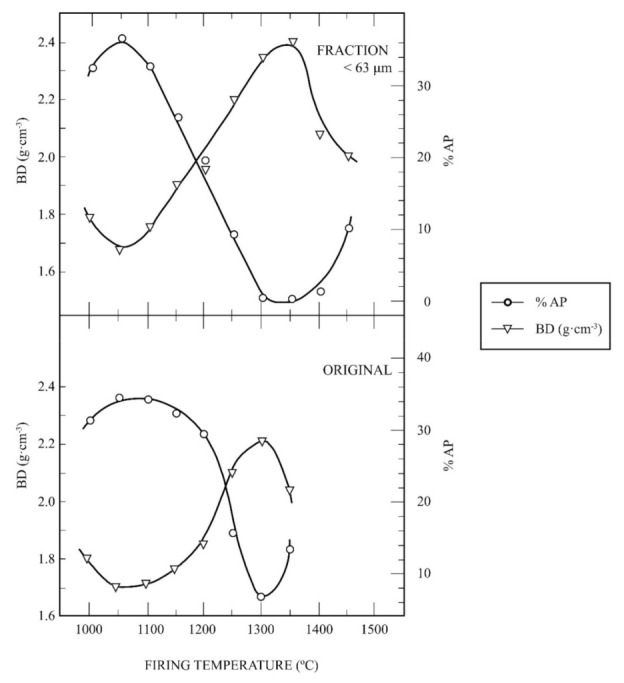
Thermal behaviour of the raw kaolin sample (labelled as ORIGINAL) and the fraction under 63 μm: plot of the variation of bulk density (BD) and open porosity (OP) as a function of firing temperature.

**Figure 10 materials-15-00583-f010:**
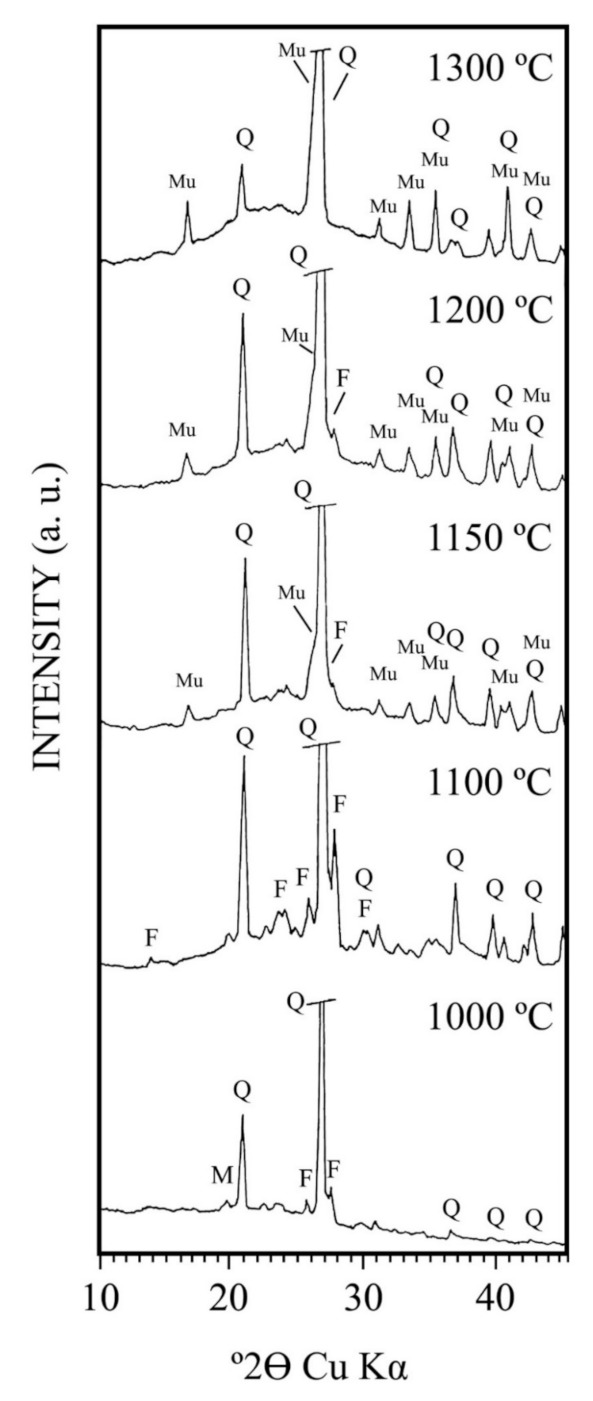
XRD diagrams of the kaolin sample (fraction under 63 μm) fired up to 1300 °C. M = muscovite (dehydroxylated); Q = quartz; F = feldspar (microcline); Mu = mullite.

**Figure 11 materials-15-00583-f011:**
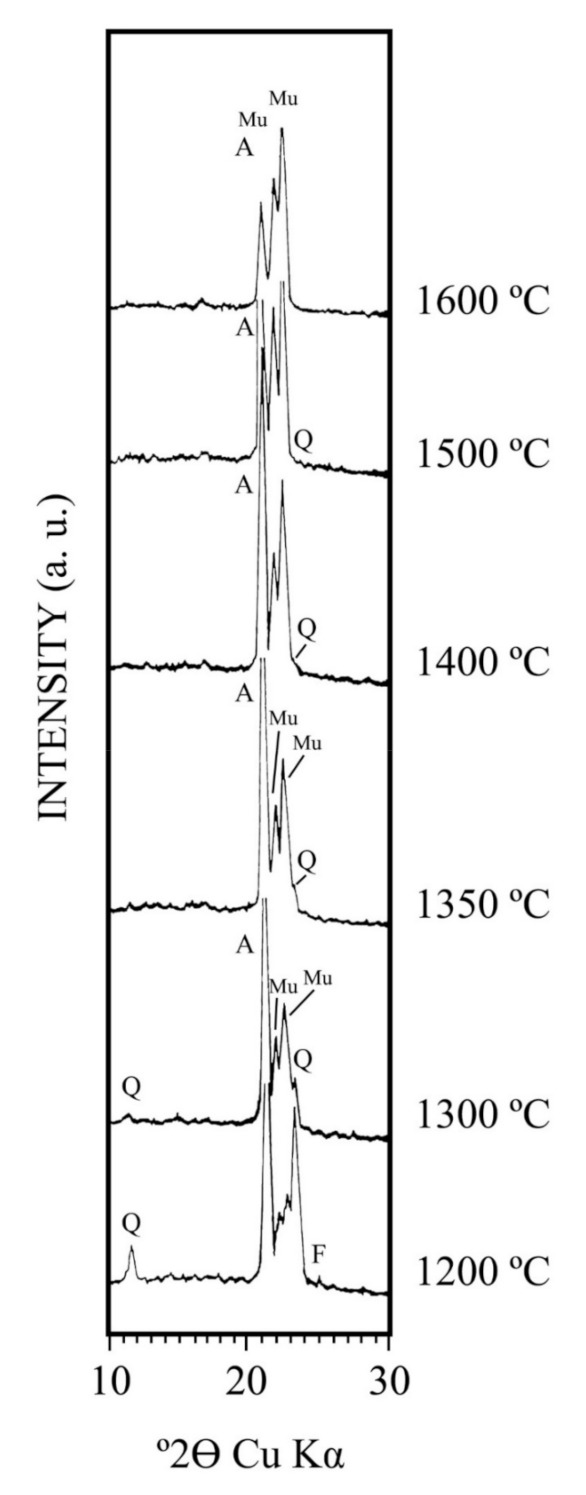
XRD diagrams of the mixture of kaolin sample (ground using a hammer mill) an α-alumina fired up to 1600 °C for 2h. F = feldspar (microcline); Q = quartz; Mu = mullite; A = α-alumina.

**Figure 12 materials-15-00583-f012:**
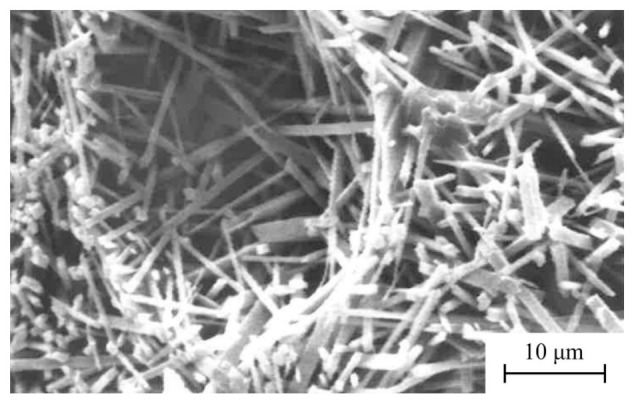
Selected SEM image of the kaolin sample (fraction under 63 μm) fired at 1300 °C for 2 h after chemical etching using 20 wt % HF aqueous solution.

**Table 1 materials-15-00583-t001:** Results of the mineralogical analysis of the samples: semiquantitative estimation of the minerals identified by XRD (in wt %).

Sample	Kaolinite	Quartz	Feldspar	Muscovite
Raw kaolin sample	20	30	35	15
Fraction under 63 µm	50	20	10	20
Kaolin of Burela	85	5	<5	10

Note: Feldspar is identified as potassium feldspar (microcline).

**Table 2 materials-15-00583-t002:** Chemical analysis by XRF of the raw kaolin sample and its fraction under 63 μm obtained by washing. It is included the chemical analysis of a sample of ceramic kaolin (washed sample) of Burela (Lugo, Spain). LOI = ”Loss on ignition” or loss of weight after heating at 1000 °C for 1 h. An estimation of the kaolinite content in the samples is calculated on the basis of the LOI results. The molar ratio [SiO_2_/Al_2_O_3_] is given for the samples.

Component (wt. %)	Raw Kaolin	Fraction under 63 µm	Kaolin of Burela	Kaolinite (Theoretical)
SiO_2_	73.61	62.17	51.15	46.5
Al_2_O_3_	13.40	23.39	33.51	39.5
Fe_2_O_3_	1.07	1.96	0.63	
TiO_2_	0.22	0.18	0.12	
CaO	0.15	0.17	0.31	
MgO	0.26	0.55	0.23	
Na_2_O	0.25	0.14	0.29	
K_2_O	8.55	4.15	1.78	
LOI	2.69	6.58	11.90	14.0
Total	100.20	99.29	99.92	100.0
Kaolinite content	~19	~47	~86	100
(SiO_2_/Al_2_O_3_)	9.33	4.52	2.59	2.00

**Table 3 materials-15-00583-t003:** Results of mineralogical and chemical characteristics (MIA and CIA parameters) deduced from mineralogical (XRD) and chemical (XRF) analyses of the samples.

Sample	MIA	CIA	Kaolinite Content (wt %) by XRD
Raw kaolin sample	46.15	59.95	20
Fraction under 63 µm	66.66	83.98	50
Kaolin of Burela	100	93.36	85

**Table 4 materials-15-00583-t004:** Experimental results of estimated vitrification temperature T_v_ (°C), temperature of the maximum apparent density T_d_ (°C) and values of maximum bulk density, BD (g cm^−3^), of raw kaolin and fraction < 63 μm studied in this work, compared to Kaolin of Burela, determined from range curves (see the text) under the same conditions.

Sample	T_v_/(°C)	T_d_/(°C)	BD/(g∙cm^−3^)
Raw Kaolin	1300	1300	2.21
Fraction < 63 μm	1300	1350	2.35
Milled kaolin sample using a hammer mill	1350	1350	2.40
Kaolin of Burela	1500	1450	2.49

## Data Availability

Data available on request due to restrictions privacy. The data presented in this study are available on request from the corresponding author. The data are not publicly available due to the restrictions of the research group on this subject.

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
