# Peer review of "Study of a Waste Kaolin as Raw Material for Mullite Ceramics and Mullite Refractories by Reaction Sintering"

_materials, 2022, doi:10.3390/ma15020583_

Round 1
Reviewer 1 Report
The research of the article has direct practical application for the beneficiation of the rock, namely, to a certain natural mine of kaolin. The paper provides a comparison with a commercial material that is quite successfully used in the ceramic industry. The work plan and experimental material are unquestionable. The manuscript was competently worked out and the research results were interpreted. However, there are a number of questions and comments that should be resolved before the publication of the manuscript.
1) It is not clear from the text (line 183) what "plaster of Paris molds" means. Will there be a difference in results if the molds are not Paris molds?
2) In the introduction, the sentence is difficult to comprehend. It needs to be modified. "The results indicated ~ 23 wt. % of particles < 63 µm and the kaolinite 18 contents of the raw kaolin and < 63 µm fraction were 20 and 50 wt. %, respectively."
3) A non-trivial representation of X-ray data, namely the representation of 2Theta from a larger angle to a smaller one. You need to motivate the choice of data presentation, or rebuild Fig. 4 to a trivial form along the abscissa axis.
4) Will the increase in the muscovite mica content in the charge affect the subsequent use in the production of ceramics? From the analysis of Table 1, the conclusion is not obvious on the enrichment of the ≤63 µm fraction with quartz, microcline and muscovite mica. The K2O content should lower the melting point and viscosity.
5) What is the error in the determination of components by X-ray fluorescence analysis (Table 2)? What standard was used for calibration in the XRF analysis?
6) Explain, what you mean by the term "iron oxide gels"? Chemically, there are iron oxides (FeO, Fe2O3 or FeO*Fe2O3) and amorphous iron hydroxide (FeO(OH)*nH2O or Fe(OH)3) or iron metahydroxide (FeO (OH)).
7) The presentation of Figures 7-9 does not seem very convenient. The legend for the symbols is difficult to identify against the background of the axis labels.
8) Why is the shrinkage of materials of the original and fraction 63 μm at 1000 ° C negative?
Author Response
COVER LETTER: Manuscript ID: Materials-1536115
This coverletter is to explain, point by point, the details of the revisions to the manuscript and the reponses to the referees’ comments.
Title: Study of a waste kaolin as raw material for mullite ceramics and mullite refractories by reaction sintering
Authors: Pedro J. Sánchez-Soto, Dolores Eliche-Quesada, Sergio Martínez-Martínez, Luis Pérez-Villarejo and Eduardo Garzón
The original manuscript has been revised following the referees’ comments and suggestions. The revised version is attached.
Changes, as recommended, have been outlined. Answers to questions, additional comments and suitable rebuttals to each reviewer are included in this coverletter.
Photographies of the deposit (1 and 2) have been included in the revised version, as suggested by a referee. Table 1 has been modified.
Figures showing the representation of X-ray data (Figures 4, 10 and 11) have been modified following the recommendations of two referees. The representation of Figures 7-9 does not seem very convenient, as indicated one referee. These Figures have been modified following this suggestion.
Figure 5 (DTA-TG) has been modified according to a recommendation of one referee. A new version is included in the revised version.
The authors made a cooperative research in Spain between engineers and chemists, members of the Department of Engineering of the University of Almería, the Department of Chemical, Environmental and Materials Engineering of the University of Jaén and the Institute of Materials Science of Sevilla, Joint Center of the Spanish National Research Council (CSIC) and the University of Sevilla. The result of this cooperative research is the paper submitted for publication in the journal “Materials”.

Reviewer 2 Report
1. Please polish the Abstract. Please add sentences to explain the meaning, the main points, the improvement and the promising application of the study.
2. It is important to describe the current progress, problems and improvements to be further studied in Introduction. Please declare the creative points. The last paragraph in Introduction should just display the major study. The sentence "It was interesting to..." is not suitable. Please correct it.
3. Please provide some practical photos about the explored materials.
4. Please number the equations in the whole manuscript. i.e. Line 341\345.
5. The major content Section 2 and Section 3 should be carefully checked. It is important to highlight the creative work and declare the meaning of the study. Improvement should be also emphsized.
6. The conclusion part should be revised. Please clearly describe the results: what has been obtained from the study and the potential use of the study.
Author Response

(The authors gave the same response as above.)

Reviewer 3 Report
Dear authors, thank you for your study, I was asked to be very brief. So, some points of concern are the following:
- The method of semi-quantitative analysis is not stated, but "was performed by the methods 167 previously described in the literature" with 5 references. Except from the fact that the majority of these 5 citations are from 3-6 decades ago, the quantitative method should be explained.
- XRD patterns. I suggest that the authors apply international standards about the presentation of their XRD patterns (corrections to x and y axes): Degrees from low to high values, addition of the missing "x" axis for B pattern in order to maintain the % in "y" axis, else replace the % with a.u.
- In the case of weight loss thermograms (figure 5), y axis is usually presented as (%).
- Please, use the same way of typing degrees (°) through the manuscript.
Author Response

(The authors gave the same response as above.)

Round 2
Reviewer 2 Report
It is suggested that the conclusions should be simplified. Some sentences can be deleted. It should just declare the major and meaningful results from the study.
Author Response
Comments and Suggestions for Authors:
It is suggested that the conclusions should be simplified. Some sentences can be deleted. It should just declare the major and meaningful results from the study.
Answer: Following these suggestions, the conclusions have been simplified in the last revised version. Some sentences have been deleted and the major and meaningful results have been declared. In fact, the name of the section ‘4. Summary and Conclusions’ in the original and revised version has been changed to ‘4. Conclusions’.
Date of this review
05 Jan 2022

Reviewer 3 Report
Dear authors, thank you for your kind response, and I'm very sorry for being so brief in the first round.
Thank you for taking under consideration all my comments.
Regarding the XRD quantitative analysis, my previous comments have to do with new proposed methods for its approach. As far as the method of quantitative analysis is now described inside the manuscript, I found it ok, as this makes things more clear for potential readers.
Thanks again for your work and your response.
Author Response
Comments and Suggestions for Authors
Dear authors, thank you for your kind response, and I’m very sorry for being so brief in the first round. Thank you for taking under consideration all my comments.
Regarding the XRD quantitative analysis, my previous comments have to do with new proposed methods for this approach. As far as the method of quantitative analysis is now described inside the manuscript, I found it ok, as this makes things more clear for potential readers.
Thanks again for your work an your response.
Answer: The interesting suggestions and comments provided by this referee in the previous report were very valuable. The XRD semiquantitative method was applied for the mineralogical analysis of the kaolin samples in this investigation. It was of utility to compare the results. The authors agree that the description of the XRD method in the revised version is more clear. Thanks a lot for your review.
Date of this review
05 Jan 2022
